# Towards a Sustainable Bioeconomy through Industrial Symbiosis: Current Situation and Perspectives

**Nicolas Bijon** [1,2,3,4,*] , **Tom Wassenaar** [1,2] , **Guillaume Junqua** [3] **and Magali Dechesne** [4]

1    CIRAD, UPR Recyclage et Risque, 34398 Montpellier, France; tom.wassenaar@cirad.fr
2    CIRAD, Recyclage et Risque, Université Montpellier, 34398 Montpellier, France
3    HSM, Université Montpellier, IMT Mines Ales, IRD, CNRS, 30100 Ales, France; Guillaume.Junqua@mines-ales.fr
4    Veolia Recherche et Innovation, Chemin de la Digue, 78600 Maisons-Laffitte, France; magali.dechesne@veolia.com
*    Correspondence: nicolas.bijon@cirad.fr

**Abstract:** The concepts of industrial symbiosis (IS) and bioeconomy (BE) both focus on ways to reduce dependence on non-renewable resources. However, these two frames of reference have rarely been considered as part of a joint strategy to achieve sustainability. Here, we describe how they inter-operate, in well documented IS case studies, to identify current synergy patterns of organic by-products, their limits, and promising pathways towards integrated initiatives that pursue the respective goals of each framework. We first evaluated the nature of synergies in current practices, and how they contribute to sustainability. Second, we focused on the role of agriculture in these symbioses, due to its fundamental role in circular bioeconomy. We used three main dimensions to analyze our case studies: IS emergence, governance of synergies, and actor serendipity. We identified three main patterns of organic matter use within IS, which we termed metabolic resources, metabolic biorefinery, and global biorefinery. Our observations suggest that synergies with agriculture are undervalued, by both internal and external practitioners. We conclude that while the combination of BE and IS can bolster sustainability, it requires a dedicated implementation strategy that has yet to be conceived.

**Keywords:** industrial symbiosis; bioeconomy; biorefinery; agriculture; typology; case studies

## 1. Introduction

As public policies increasingly support the transition towards sustainability [1–3], it is crucial to examine the actual contribution of collective action paradigms, and the ways in which they can reinforce one another. In particular, the normative action plan of Circular Economy (CE) [4–6] and the concept of bioeconomy (BE) [7,8] are receiving increasing attention from both academia and decision makers. Recent studies described how the diversity of sustainability frameworks contributes to sustainability at different levels [9,10]. Circular bioeconomy has emerged as a combined approach to BE and CE [11,12], but is more than a mere combination. These apparently distinct frameworks share the potential to implement and develop local collective actions [13–16], such as those exemplified by industrial symbiosis (IS) initiatives [17–19]. Biorefineries are the most representative examples of the combination of BE and CE through local action. Ubando et al. [20] define the term biorefinery as "an infrastructure facility wherein various conversion technologies such as thermochemical, biochemical, combustion, and microorganism growth platform are integrated to efficiently produce sustainable bio-based product streams such as bio-fuels, biochemicals, bioenergy, and other high-valued bio-products". Only a few authors have considered IS as a means to reach the goals of BE, mainly from a biorefinery stand-point [21,22]. Yet a broad inventory and analysis of such practices is lacking, to date, even though evidence suggests that organic matter (OM) can be integrated into IS in more

diversified forms [23,24], thus creating a potential increasing in the recycling of organic matter. However, the questions of how, and to what extent, industrial symbiosis addresses bioeconomic goals, as well as whether, and how, this situation might be improved, remain to be addressed. These questions are of great importance to researchers and practitioners, as they will allow identification of the best opportunities to contribute efficiently to sustainability through the implementation and development of IS. The aim of our study is to advance current understanding by providing theoretical and descriptive tools to interpret the place of BE in IS case studies, and by identifying opportunities for future development of IS initiatives dedicated to BE.

In Section 1, we present the different visions of BE, based on which we developed our analytical framework, and detail the objectives of our work; in Section 2, we describe the methodology we used to assess the features of BE and characterize the IS case studies; in Section 3, we review 30 selected IS case studies to understand the role of BE within them; and in Section 4, we suggest future opportunities based on our observations.

### 1.1. Visions of Bioeconomy

The concept of BE covers a broad and still-debated range of strategies that incorporate different approaches to sustainability [25]. Bugge et al. point out that BE actually embraces three contrasting visions: bio-technology, bio-resources, and bio-ecology [7,26].

The bio-technological vision of bioeconomy (BT-BE) emphasizes opportunities for innovation and economic growth by expanding the application range of technologies involving biological processes or living beings. These technologies include a wide range of fields, including genetics, genomics, GMOs, biofuels, nano-biotechnologies, and medicine. Environmental benefits are expected to be one consequence of this strategy, which relies, to a great extent, on future technology to address present issues [27,28].

The bio-resource vision of bioeconomy (BR-BE) aims at expanding the range of uses of biological resources, by means of new processing technologies, to increase the added value of these materials while replacing fossil-based inputs [29]. This vision builds on the idea of "cascading biomass" [30,31] that prioritizes the use of OM according to its potential value, including organic waste. This vision of BE also depends, to a great extent, on scientific innovation, as many processing technologies are still insufficiently mature [22]. The BE core concept of biorefinery is included in this vision [20,32–34].

The bio-ecology vision of bioeconomy (BE-BE) highlights the importance of material flows, stemming from biomass in the regional and global metabolisms, to achieve sustainability [35]. Such an approach requires consideration of the impact of land footprints and biocapacity [36]. It also requires regional governance specifically designed to address this issue [13]. This vision of BE can be linked with the historic use of the term bioeconomy in Georgescu-Roegen's 'The Entropy Law and the Economic Process' [37].

### 1.2. Objectives of Our Work

The concept of strong sustainability assumes physical and biological limits to the development of human society [38,39], as opposed to the notion of weak sustainability, which assumes that the depletion of natural capital can eventually be offset by increasing human-made capital [40]. This distinction introduces a normative hierarchy among so-called sustainable practices, as, according to some authors [41] weak sustainability practices actually provide no guarantee of sustainability—a viewpoint with which we agree. This is why we consider compatibility with strong sustainability to be an interesting qualitative assessment tool for synergy patterns. Such an approach has already been used to assess policies and practices linked to sustainable development [42].

In this paper, we define industrial symbioses including bioeconomy (BE-IS), as initiatives which include at least one exchange of organic byproducts. The objective of our work was to assess the extent to which BE-ISs contribute to strong sustainability, and to the opportunity to strengthen this contribution. To this end, we analyzed the following aspects: (1) the characteristics of BE-IS and its frequency among existing IS case studies;

(2) the underlying principles of OM synergy and its link to BE visions, which we term 'synergy patterns'; and (3) the role of agriculture as a cornerstone of sustainable bioeconomy. The following sections (Sections 1.2.1–1.2.3) contextualize and specify these three objectives, which structure the Results and Discussion sections (respectively, Sections 3 and 4).

### 1.2.1. Identifying the Characteristics of BE-ISs

In theory, BE can be performed through ISs, even though this is not the primary objective. Indeed, ISs are dedicated to all types of by-products, without a particular focus on OM. Inversely, BE traditionally considers all sources of OM, and is not limited to by-product streams. Recent contributions suggest that BE could be associated with CE to improve their respective sustainability through circular bioeconomy [11,12]. ISs are tools which could allow the development and implementation of this new framework, as they already do for CE [19]. In some IS case studies, for instance, OM is already a central synergistic by-product [23,43]. Some authors have highlighted the importance of including organic waste in BE, as it has high potential for resource recovery [44,45]. For instance, Venkata Mohan et al. [46] underline the potential contributions of organic waste to biorefineries, a practice that falls within the range of IS.

Despite these contributions, the actual place of BE within existing IS diversity has not yet been documented. Moreover, the uses of OM inside ISs, and the actors related to these uses, have not been thoroughly described, and therefore remain unclear. Our first objective is to describe these features in a selection of well-documented case studies, and to compare the characteristics of BE-ISs with IS case studies that do not include synergies with organic by-products.

### 1.2.2. Identifying Synergy Patterns and Their Link to Sustainability

As the following overview suggests, BE and IS both contribute, incompletely, to sustainability [9].

IS aims at enhancing regional collaboration between economic actors to improve environmental and economic efficiency [18,47]. Synergies between different actors allow one's waste to be used as an input for another [48], thereby reducing the consumption of raw non-renewable material. Industrial ecology principles promote strong sustainability [42,49], although practical applications more likely involve weak sustainability, namely due to their transfer in the regulative framework of the circular economy [3,4,10].

The contribution of BE to sustainability is hotly debated, since it depends on the vision concerned [50]. The BE-BE vision is explicitly dedicated to strong sustainability, as it focuses on the regional metabolism and its biological and physical capacity. The two other visions, although not incompatible with strong sustainability, endorse weak sustainability, as they promote technological development without a thorough consideration of its limits. The two other visions are also the main focus of the European BE strategy, which is defined as "all sectors and systems that rely on biological resources (animals, plants, micro-organisms and derived biomass, including organic waste), their functions and principles." [1] (p. 4). This institutional vision of BE suggests an increase in technological innovation to foster economic growth and increase the use of biomass resources in all economic sectors [25,29], which is likely to result in generally weak sustainability [10,36,42,51].

In summary, IS is dedicated to long-term circularity, but maintains the current productive system, thereby continuing to cause environmental degradation and resource depletion. BE aims to bring about long-term changes in the production system in spite of its actual institutional framework, which promotes weak sustainability. It is thus of interest to assess which type of BE, integrated into IS, would best enhance sustainability. Based on an in-depth analysis of OM synergies within BE-IS, our second objective is to assess how these synergies relate to strong sustainability by considering the three visions of BE (see Section 1.1). To this end, we introduced the concept of synergy patterns, i.e., families of synergies that exhibit similar visions of BE.

1.2.3. Assessing the Role of Agriculture in IS

The sustainability of agriculture is a major concern for the coming decades. One of the main challenges is to reduce the dependence of the agricultural sector on non-renewable resources [52,53]. The long-term provision of nutrients, such as nitrogen and phosphorus, is a major challenge for food production [54]. Nitrogen fertilizers are largely sourced from non-renewable resources, and are an important outlet for fossil fuels not used in energy production [55]. In theory, IS enables these non-renewable inputs to be replaced through regional collaboration. Evidence abounds that many of the nutrients required for agriculture could be sourced from organic waste, especially in proximity to dense urban areas [56,57]. Nevertheless, organic waste remains poorly valorized [58]. Despite the widespread recovery of organic waste in agriculture, recovering nutrients from those sources still has high potential [59]. As such, agriculture has a high potential to both accept flows from many different OM inputs and produce a large volume of waste or by-products [35]. These two characteristics define the role of an anchor tenant in IS [60], a role which could, theoretically, be held by agriculture in some situations. Such practices are rarely covered in IS literature, even though bridging this gap could increase nutrient circularity and overcome this recurring blind spot [61]. Fernandez-Mena et al. [24] examined the potential of IS tools to foster synergies in the agricultural sector. However, the role of agriculture in current IS initiatives has not yet been investigated. Given the fundamental importance of sustainable biomass production, and the potential of IS to contribute to this objective, our third objective is to assess the role of agriculture in the selected IS case studies.

**2. Materials and Methods**

We followed four main methodological steps for this study, shown in Figure 1: (1) Selecting the case studies; (2) identifying a relevant analytical framework incorporating IE and BE characteristics; (3) describing the case studies using this framework; and (4) analyzing BE-IS according to the objectives of this study.

**Figure 1.** Methodological approach of this study.

*2.1. Selection of the Case Studies*

For our study, we needed cases that were sufficiently documented in the literature (our main selection criterion), and that covered a diversity of situations and contexts. We excluded cases that were only documented in non-academic literature. Our set of case

studies was based on the combined sets of IS cases reviewed in two recent articles which placed particular emphasis on selecting sets of cases that correctly represented the diversity of situations and contexts in which IS has recently developed. Boons et al. [62] developed an IS typology based on a careful selection of case studies, while Mortensen and Kørnøv [63] selected case studies to assess the conditions of IS emergence. Among the case studies cited in these two reviews, we selected those in which sufficient and reliable data informed all the IS characteristics covered in the present article (Figure 1): the three dimensions of our typology (Section 2.2.2), the presence of anchor tenants, industrial diversity, spatial range, and material flows. To this end, and when needed, we included other papers reporting on the same case studies, either cited by [62,63] or found in Google Scholar searches. This resulted in 20 case studies taken from the review by Boons et al. [62], and an additional 10 case studies from the review by Mortensen and Kørnøv [63]. The selected case studies are listed in Table 1 (for more details see Appendix C).

**Table 1.** Main characteristics of the case studies selected. An extended table is provided in Appendix C. Country names are based on ISO 3166. Abbreviations used for continents: AS, Asia; EU, Europe; NA, North America; OC, Oceania; SA, South America. Unk. indicates an unknown starting date. Emergence, Governance, and Serendipity are defined in Section 2.2.2.

| Name, Country, Continent, and Starting Date | Corpus | Emergence | Governance | Serendipity | BE-IS | Refs. |
|---|---|---|---|---|---|---|
| Kalundborg, DK, EU, 1959 | [62] | Internal | Self-organized | Goal-Directed | Yes | [64–66] |
| Styria, AT, EU, Unk. | [62] | Internal | Self-organized | Serendipitous | Yes | [67] |
| Guayama, PR, NA, 1990's | [62] | Internal | Self-organized | Goal-Directed | No | [68] |
| Kwinana, AU, OC; 1952 | [62] | Internal | Facilitated | Goal-Directed | Yes | [69,70] |
| Gladstone, AU, OC, 1967 | [62] | Internal | Facilitated | Serendipitous | No | [70] |
| Nanjangud, IN, AS, Unk. | [62] | Internal | Self-organized | Serendipitous | Yes | [71] |
| Jyvälskylä, FI, EU, Unk. | [62] | Internal | Self-organized | Goal-Directed | Yes | [60] |
| Kuusankoski, FI, EU, 1880's | [62] | Internal | Self-organized | Serendipitous | Yes | [72,73] |
| Guitang Group, CH, AS, 1956 | [62] | Internal | Self-organized | Goal-Directed | Yes | [74,75] |
| British Sugar, GB, EU, 1985 | [62] | Internal | Self-organized | Serendipitous | Yes | [76] |
| HISP, GB, EU, 2000 | [62] | Hybrid | Facilitated | Goal-Directed | Yes | [77] |
| WISP, GB, EU, 2000. | [62] | Hybrid | Facilitated | Serendipitous | Yes | [77] |
| Rotterdam, NL, EU, 1994 | [62] | Internal | Facilitated | Goal-Directed | No | [78,79] |
| Ulsan, KR, AS, 1990 | [62] | External | Facilitated | Goal-Directed | No | [80,81] |
| TEDA, CH, AS, 2000 | [62] | External | Facilitated | Goal-Directed | No | [82,83] |
| Fort Devens Army Base, US, NA, 1993 | [62] | External | Facilitated | Goal-Directed | No | [84] |
| Händelö island, SE, EU, Unk. | [62] | Internal | Facilitated | Serendipitous | Yes | [85,86] |
| Biopark Terneuzen, NL, EU, 1998 | [62] | External | Facilitated | Goal-Directed | Yes | [87] |
| SYIA, CN, AS, 1998 | [62] | Internal | Self-organized | Serendipitous | No | [88,89] |
| Campbell, US, Hawai, 1992 | [62] | Internal | Self-organized | Serendipitous | No | [90] |
| Porto Marghera, IT, EU, 1970's | [63] | Hybrid | Facilitated | Goal-Directed | No | [91] |
| Industrial Symbiosis Platform (Sicily), IT, EU, 2011 | [63] | Hybrid | Facilitated | Serendipitous | NA | [92] |
| Relvão EIP, PT, EU, 2004 | [63] | Hybrid | Facilitated | Goal-Directed | Yes | [93] |
| Barceloneta, PR, NA, 1970's | [63] | Internal | Self-organized | Goal-Directed | Yes | [94] |
| NRIA, TH, AS, 2000 | [63] | External | Planned | Goal-Directed | NA | [95] |
| Santa Cruz EIP, BR, SA, 2002 | [63] | External | Facilitated | Goal-Directed | No | [96,97] |
| Paracambi, BR, SA, 2006 | [63] | External | Planned | Goal-Directed | No | [96] |
| Bazancourt-Pomacles, FR, EU, 1990 | [63] | Internal | Facilitated | Goal-Directed | Yes | [98] |
| Kawasaki, JP, AS, 1997 | [63] | External | Facilitated | Goal-Directed | Yes | [99,100] |
| Deux Synthe, FR, EU, 2000 | [63] | Hybrid | Facilitated | Goal-Directed | No | [101] |

In the most recent comprehensive review, Neves et al. [17] identified a total of 124 case studies referenced in the literature. Our set of 30 case studies corresponds to an important section of these case studies, for which the required data were available. We consider them to be representative of this set, knowing that they were originally selected by their authors to cover a diversity of situations. It is important to note that BE considerations or material exchanges were not criteria for the selection of our case studies. To avoid bias, and to identify all existing practices in a representative set of ISs, we chose to identify which IS integrate OM in their synergies ex-post (BE-ISs, see Section 1.2). The selected ISs are, consequently, precious examples that inform how features that now fall within the BE framework were already integrated in these initiatives. From these data, we were able to identify the most common practices representing synergies that incorporated organic matter, without restricting our scope.

### 2.2. Analytical Framework

To identify the relevant analytical framework, we drew a structured list with a large number of potentially distinguishing characteristics, completed using available quantitative or qualitative information. The qualitative use of literature data sometimes required interpretation, which we attempted to minimize by using clear definitions. To avoid the problem of the potential differences between the description of case studies in the literature and their current status—which may not be easily accessible—we used the material as if the data were still up to date. This interpretation of the first set of case studies was compared with additional cases and improved in the second step. Finally, the most relevant characteristics, applicable to all case studies, were selected to validate the descriptive framework we used for our analysis. We present these characteristics, including our three-dimensional typology, in the following sections.

#### 2.2.1. Assessment of BE and IS Characteristics

For each case study, we determined a set of characteristics that enabled us to understand its relation to BE. We assessed (1) the industrial diversity based on NACE typology [102], to evaluate the complexity of the network of synergies and the type of actors present; (2) the presence of an anchor tenant structure [60], including the sector of the anchor tenant, when relevant, to determine if actors were predominant—and if so, which ones; and (3) the spatial range, which we assessed on a qualitative scale due to the difficulty assessing this point precisely, to evaluate the importance of spatial distance. More details on the assessment method for each of these points are provided in Appendix A.

For BE-ISs, we (1) analyzed the actors involved in all known OM synergies, (2) described these synergies, to derive the different types of OM use, and (3) analyzed the role of agriculture in order to understand how agriculture interacts with other actors. From this description of synergies, we inferred the visions of BE to which they corresponded (see Section 1.1) and grouped them under general patterns, i.e., synergies that share the same visions. This allowed us to differentiate the synergies within BE-ISs, and to discuss their respective contribution to sustainable BE.

#### 2.2.2. Classification: Building a Typology of IS through Constitutive Dimensions

We used a typology of IS to compare the above-mentioned characteristics. To this end, we described the nature of each case study as a combination of three constitutive dimensions recognized in the literature. We noted that the terms 'self-organized', 'facilitated', and 'planned', formalized by Chertow (2007) [103], can refer to either the course of the IS [104] or to the emergence process [86]. For this reason, we decided to separate the two components in the description of our case studies. To refer to emergence, we differentiate symbioses initiated by (i) internal actors, (ii) external actors, and (iii) both internal and external actors [93]. We refer to the governance of existing initiatives using the standard terms (i) self-organized, (ii) facilitated, and (iii) planned [103]. To this, we added a third dimension, namely the mode of identification of new material synergies, using Paquin & Howard-

Grenville's distinction between (i) serendipitous and (ii) goal-directed synergies [105]. We argue that, for our analysis, these dimensions should not be reduced to one, which could cause confusion in time scales and IS objects, even though they are not totally independent. We assessed these dimensions using rigorous questions, and separated the case studies into, respectively, three, three, and two sub-categories, as shown in Figure 2. These dimensions theoretically form 18 combinations, or "IS types", which we designated with trigram codes composed of the initial of each dimension (Figure 2), in the following order: emergence, governance, and serendipity. For instance, "IFS" refers to a symbiosis with internal emergence, facilitated governance of synergies, and serendipitous goals for the actors. We use the letter x to refer to any sub-category of a missing dimension. Due to dependencies among the three sub-categories, the following combinations cannot occur:

Emergence processes in which actors are involved (Internal and Hybrid) exclude planned (non-participatory) governance for synergy development; thus, IPx types and HPx types (IPS, IPG, HPS, HPG) are excluded.

Self-organization of synergies is only possible with internal emergence; thus, HSx and ESx types (HSS, HSG, ESS, ESG) are excluded.

Planned governance excludes serendipity; thus, EPS is excluded.

These dependencies leave nine possible types: ISS, ISG, IFS, IFG, HFS, HFG, EFS, EFG, and EPG. Details on these categories and dimensions, as well as definitions, are provided in Appendix B.

| Dimension | Emergence |
|---|---|
| Question addressed | Who is the initiator of the IS? |
| Object | Symbiosis |
| Time Scale | Beginning of IS |

| Sub-categories | Brief definition |
|---|---|
| I*xx* / Internal | IS engages spontaneously |
| H*xx* / Hybrid | IS comes from internal and external contributions |
| E*xx* / External | IS is driven by an external actor |

| Dimension | Governance |
|---|---|
| Question addressed | How are the synergies designed and agreed upon? |
| Object | Synergies |
| Time Scale | Current state |

| Sub-categories | Brief definition |
|---|---|
| *x*S*x* / Self-Organized | Synergies are bilateral contracts |
| *x*F*x* / Facilitated | Synergies emerge from a collective process |
| *x*P*x* / Planned | Synergies are engineered externally |

| Dimension | Serendipity |
|---|---|
| Question addressed | What are the synergies for? |
| Object | Actors |
| Time Scale | Future goals |

| Sub-categories | Brief definition |
|---|---|
| *xx*S Serendipitous | Synergies serve actors' interests |
| *xx*G Goal-Directed | Synergies serve an overarching interest |

**Figure 2.** Description of industrial symbiosis through a 3-dimensional typology. See Appendix B for more details on the different dimensions.

### 3. Results

*3.1. Frequency and Characteristics of BE-IS among Existing IS*

Figure 3 presents the characteristics we assessed to compare BE-ISs to other ISs. The notable variation among industrial symbioses indicates that our selection methodology successfully provided a diverse sample of case studies. Table 1 maps the selected case studies according to the typology used.

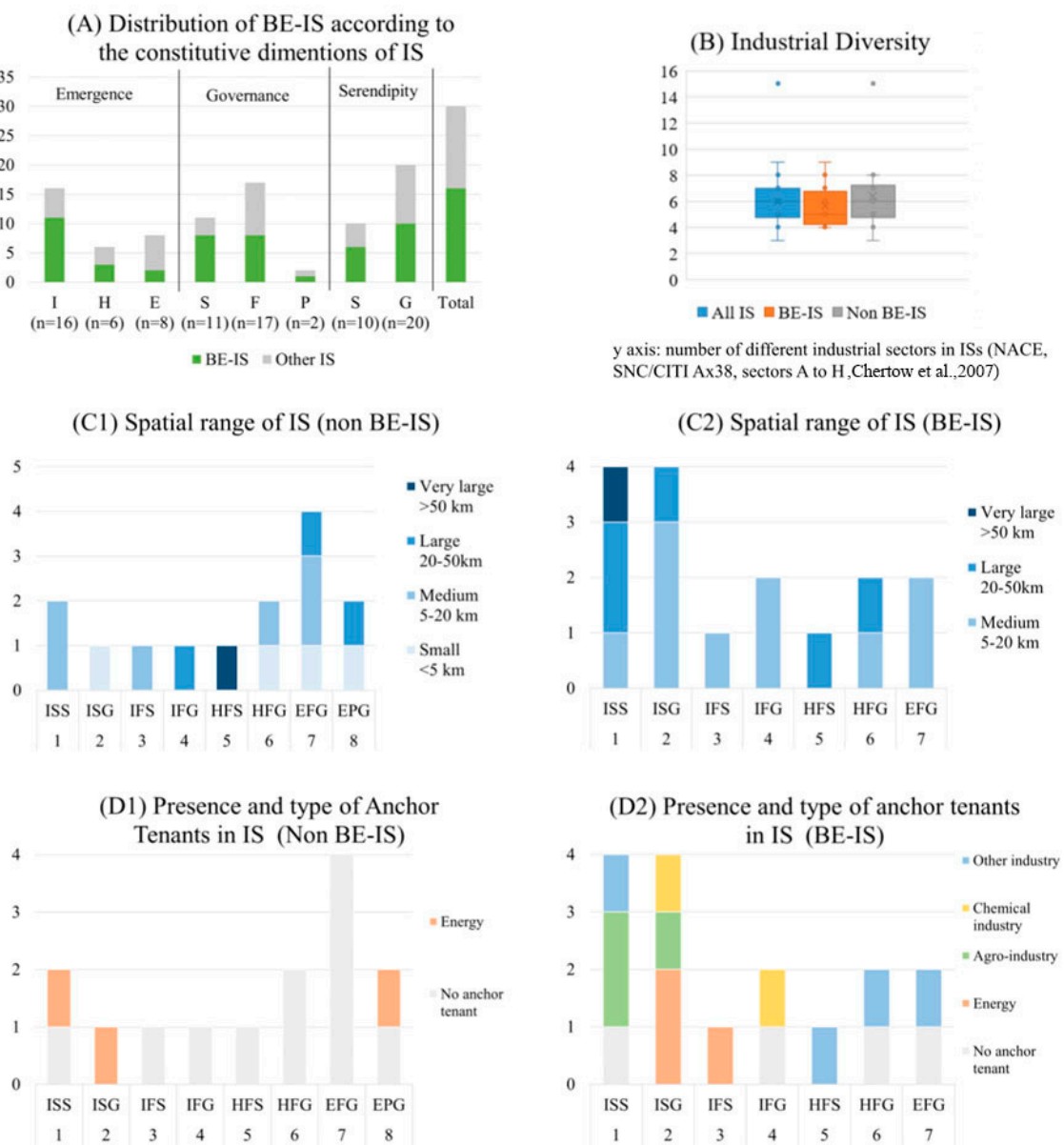

**Figure 3.** Characteristics of the selected ISs according to the typology, based on three constitutive dimensions: distribution of the BE-ISs (as defined in Section 1.2) (**A**); comparison of the industrial diversity of all selected ISs, BE-ISs, and non BE-ISs (**B**); Spatial range of non BE-ISs (**C1**) and BE-ISs (**C2**); Anchor tenants in non BE-ISs (**D1**) and BE-ISs (**D2**) The *Y* axis in all panels (except (**B**)) corresponds to the number of observations among the selected case studies. Abbreviations and trigrams are defined in Section 2.2.2 and in Figure 2. The type of anchor tenants refer to NACE categories (see Appendix A). BE-IS = IS implies at least one exchange of organic matter (cf. Section 2.2). For more information, see the methodology in Section 2.

The case studies fall into eight out of the nine expected types, as EFS was not observed. We cannot conclude if the absence of this type stems from the impossibility of serendipity in the context of external emergence, or if such a case, though rare, actually exists. The most common type of emergence among the selected case studies was internal emergence (53%), the most common type of governance was facilitation (57%), and 66% of the cases were goal-directed. Industrial diversity ranged from 3 to 11 different industrial sectors, with a median value of six and a mean deviation of 1.37 (Figure 3B). Fifty percent of the IS case studies (15 cases) showed an anchor tenant structure. In these cases, the industrial sector that held the role of anchor tenant most frequently was energy (Category D in NACE;

six cases). Other frequent anchor tenants were agro-industry (Category CA; three cases) and chemical industry (including pharmaceutics and refinery, categories CD, CE and CF; three cases). Anchor tenants were more frequently represented in ISx types. Thirteen percent of symbioses were within a small spatial range (<5 km), 53% in a medium range (5–20 km), 27% in a large range (20–50 km), and 7% in a very large range (>50 km). The maximum range observed was around 200 km.

Among the 30 selected case studies, 16 were considered BE-ISs, i.e., ISs that include at least one exchange of an organic by-product (Figure 3A). In two cases, the available data did not allow us to conclude whether such exchanges had taken place or not. The industrial diversity of these 16 BE-ISs was 5.7, with a standard deviation of 1.27, which was not significantly different from the calculated industrial diversity of the whole set of case studies (Figure 3B). We calculated that 63% of BE-ISs (10 cases) were in a medium spatial range, 31% (5 cases) in a large range, and the last case in a very large range. No BE-IS occurred in a small spatial range (Figure 3C2). Seventy-five percent of BE-ISs had an anchor tenant structure. This proportion was higher than that of all case studies, especially Hxx and Exx types. Most of the ISs with an anchor tenant actually correspond to BE-ISs (12/15), including seven types of possible anchor tenant sectors (among which agro-industry, chemical industry, and the energy sector were the most frequently represented) (Figure 3D2).

The 14 remaining non-BE-ISs exhibited contrasted features. Concerning the spatial ranges (Figure 3C1), six ISs were in a medium range (43%), four in a small range (29%), three in a large range (21%), and one in a very large range (7%). Overall, the non-BE-ISs were, thus, generally smaller than the BE-IS. Only three non-BE-ISs had an anchor tenant (21%), all in the energy sector (Figure 3D1). This contrasts with the high frequency and diversity of anchor tenants in BE-ISs.

*3.2. Patterns of Bioeconomic Synergies in BE-ISs*

3.2.1. OM Actors, Uses, and Synergies

OM actors in BE-ISs fall into six main NACE categories: (i) agriculture (category A of NACE classification), including crop and livestock farming, forestry, and fishery; (ii) agro-industry (category CA), including production of food, drink, and tobacco products; (iii) wood industry (category CC), including processing of wood, paper, and carton products; (iv) chemical industry (category CE), including the refinement of organic products to obtain alcohol or biofuels, as well as the extraction of organic molecules; (v) energy production (category D); and (vi) waste management (category E), including anaerobic digestion. Other industry sectors were also present, but to a lesser extent, e.g., CD (refinery industry) and CG (plastic, stone, and ceramic industries).

Figure 4 shows the synergies observed in the case studies according to the actors involved and the uses of OM. Among these actors, we derived six types of OM uses, based on the synergies shown in Figure 4: (i) direct agricultural use; (ii) direct energetic use; (iii) use as raw material for manufacture; (iv) processing into another product ultimately destined for (iv-a) (indirect) agricultural use, (iv-b) (indirect) energetic use, or (iv-c) other products.

Synergies in which agriculture is the receiver of an organic by-product or waste can be considered a direct agricultural use of OM. These include the spreading of by-products, such as compost (British Sugar) or sludge (Kwinana), the use of unprocessed by-products as animal feed (Handelö) or insect feed (Kwinana), and mulching (HISP). Indirect agricultural uses refer to processing of by-products or waste in which the processed product is destined for use in agriculture. These include the production of animal feed (British Sugar) and fertilizer (Relvão EIP, Kalundborg). This category does not include products that could be considered waste by-products of processing in which agricultural recycling is not the primary goal—such synergies would fall under direct use. Direct use of OM for energy production implies its combustion to produce heat or electricity, which can either be used in the production process of an actor (Nanjangud) or distributed to other actors through an

electrical grid (Kuusankoski). Indirect energy use corresponds to the production of energy carriers, such as biogas, bio-diesel (HISP), or alcohol (Terneuzen, Bazancourt-Pomacle). Raw material use corresponds to situations in which a manufacturing process includes OM for its intrinsic characteristics, as is the case in the production of paper (Styria), cement (Guitang), and pet food (HISP). This differs from processing into other products, in which organic by-products serve as a basis, on their own, for the creation of products, with value added through green chemistry (Bazancourt-Pomacle), separation processes (British Sugar), or oil extraction (Nanjangud).

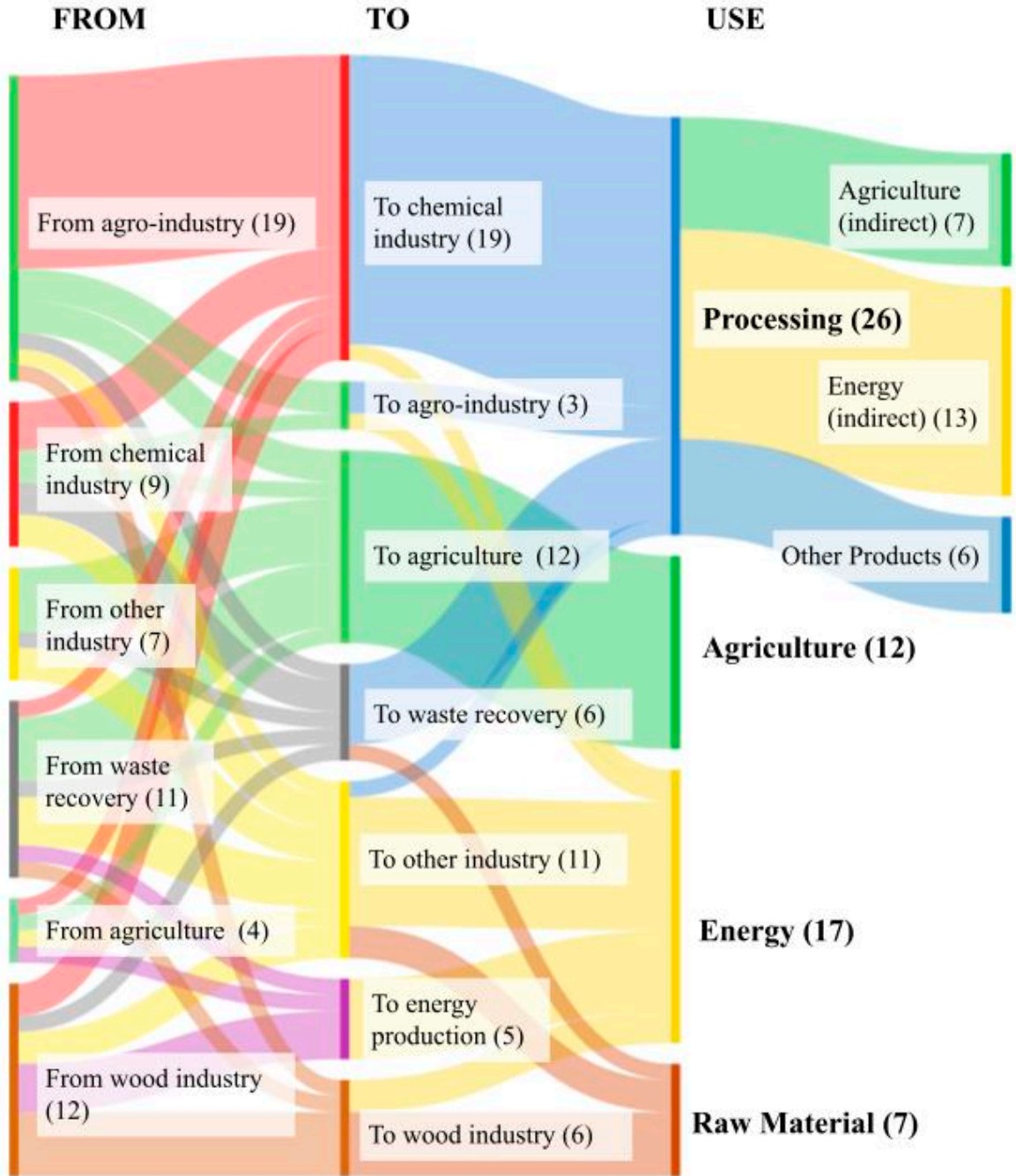

**Figure 4.** Origin, destination, and use of organic by-products in synergies involving these products in the selected BE-ISs. The first column lists the producer of the by-product; the second column lists the receiver of the by-product; the third column lists the use of the by-product by the receiver; the fourth column lists the end purpose of the processed product, when relevant. Numbers indicate the relevant number of flows in all case studies combined.

Most of these synergies involve the processing of residual OM (indirect use). Energy provision was the main purpose of synergies, corresponding to 37% of OM use occurrences, with 27% being direct use, performed by diverse types of industries. Energy provision was followed by processing into other products (32%) and agriculture (32%, with 21% corresponding to direct use). When manufactured products contributed indirectly to agricultural or energetic use (e.g., fertilizers and biofuels, respectively), we were generally unable to assess whether their use occurred inside or outside the perimeter of the ISs. We observed many exchanges between agro-industries and chemical industries related to the processing of OM. Another important actor in processing was the waste industry, which accepts organic by-products from all types of actors. We observed some cases of within-sector valorization, i.e., recycling, within the wood industry (for instance, paper production in Styria), the chemical industry (resin separation and fermentation processes in chemical operations by British Sugar), the agro-industry (pet food made from food waste in HISP), and agriculture (farm composting and spreading in horticulture, linked to British Sugar).

3.2.2. Description of Synergy Patterns and Their Place among BE-ISs

The term 'synergy patterns' groups synergies that share a similar vision of BE (as discussed in Section 1.1). Our first observation was that no synergy could be linked to a single vision of BE. Thus, the identified patterns are a combination of visions. Different patterns can co-exist within initiatives.

A wide range of practices aim to use local by-products as direct inputs, and we termed this pattern the *metabolic resource* pattern. This pattern is based on the potential to extend the use of OM to contribute to actors' local needs, directly in line with the original goal of IS. Here, we recognize the BR-BE and BE-BE visions of BE. All direct uses of OM as matter, energy, or agricultural inputs or are part of this pattern. This pattern covers situations in which agriculture provides OM to an anchor tenant industry, either as an input for processing, as a by-product (British Sugar) or as an energy carrier (Jyvälskylä, Kuusankoski). Sometimes, by-products, such as slurry or sludge, are received from other local industries (Kalundborg, Kwinana). This pattern also covers the incorporation of OM in processes in which it replaces raw material (Guitang, Styria), as well as the opportunity to use co-located organic by-products for energy production (Kawasaki, Relvão). In our series of case studies, we observed a predominance of the direct use of OM (59% of direct uses) for energy production.

The second pattern we identified was the transformation of local organic by-products into higher value products. This actually corresponds to the definition that Ubando et al. used for biorefinery (Ubando et al., 2020; see Section 1.1), which is in line with the purpose of BE. This pattern combines the BR-BE and BT-BE visions, as it relies on processing technologies to increase the exploitation of organic resources. This is mainly the case for energy carriers, such as alcohol (Terneuzen), biogas (Kalundborg), and biofuels (HISP). Some ISs also produce fertilizer (Relvão) or animal feed (British Sugar) whose use is not explicitly stated to be local. This pattern also includes green chemistry products (Bazancourt). In most cases, these products leave the perimeter of the ISs to join the global market, which is why we call this pattern *global biorefinery*. On the other hand, there are ISs in which processed products return, at least partially, to local actors. In this case, the use of a processed product as a local resource also fits the BE-BE vision. We thus derived a third synergy pattern that incorporates all three visions: *metabolic biorefinery*. Processed products clearly have the potential to replace raw materials, especially if they are designed to meet the local requirements of other actors, including the local use of fertilizers (Kalundborg, Guitang, Handelö) or energetic products (Handelö).

Based on these definitions, we identified the types of IS in which these patterns were present in our set of case studies (Table 2). The metabolic resource pattern appeared in most of the BE-ISs (14/16); namely, in all case studies with internal emergence or self-organized synergies. This pattern appears to be the most frequent and spontaneous way to integrate

OM into synergies. Global biorefinery (9/16) was more frequent when external actors were involved, or when there was a facilitation process. Metabolic biorefinery was the least common pattern (6/16), and was slightly more present in ISs with internal emergence, self-organized synergies, and serendipitous actors. Table 2 also presents the main assets and limits of these patterns in contributing to strong sustainability, related to the role of agriculture in the symbiosis.

**Table 2.** Presentation of BE-IS patterns, their contribution—or the threats they represent—to sustainable biomass production, and their occurrence among the case studies. As explained in Section 1.1, BT-BE, BR-BE, and BE-BE refer to bioeconomy visions. Percentages indicate the proportion of ISs showing at least one of the corresponding patterns, according to a specific dimension. For instance, 100% of IS with internal emergence (*n* = 11) contain synergies that correspond to the metabolic resource pattern, while only 45% include synergies with a metabolic biorefinery pattern.

| **Pattern** | | Metabolic Resource | Metabolic Biorefinery | Global Biorefinery |
|---|---|---|---|---|
| **Uses of OM** | | Direct agricultural use, direct energy use, raw material. | Indirect use of OM, with a recipient explicitly located inside BE-IS boundaries. | Indirect use with recipients presumed to be outside BE-IS boundaries. |
| **Visions of BE** | | BR-BE and BE-BE. | BT-BE, BR-BE, and BE-BE. | BT-BE and BR-BE. |
| **Levers for strong sustainability and sustainable agriculture** | | Cheap recycling of nutrients. Agriculture as a resource user. | Locally adapted nutrient recycling. Agriculture defines a specific need. | Through global recycling of nutrients. Alternative production of industrial fertilizer. |
| **Limits to strong sustainability and sustainable agriculture** | | Priority given to direct energy use. Agriculture as an outfall for waste. | Drift towards global biorefinery due to the priority given to the most valuable use. Agriculture as a provider of raw material. | Priority given to the most valuable use. Agriculture as a provider of raw material Agriculture not involved in synergies. |
| **Presence of the pattern in BE-ISs (*n* = 16)** | | 88% | 38% | 56% |
| **Presence according to SI emergence** | Internal (*n* = 11) | 100% | 45% | 45% |
| | Hybrid (*n* = 3) | 67% | 33% | 100% |
| | External (*n* = 2) | 50% | 0% | 50% |
| **Presence according to synergy governance** | Self-Organized (*n* = 8) | 100% | 50% | 50% |
| | Facilitated (*n* = 8) | 75% | 25% | 63% |

*3.3. Role of Agriculture*

Agriculture was present in most of the BE-ISs (14/16). Our first observation was that agriculture as an actor is poorly documented, and generally appears at the periphery. In the included case studies, authors often describe agriculture very approximately, as 'nearby farms' [64], 'local farmers' [71], or simply 'agriculture' [72]. This vagueness makes it impossible to evaluate whether synergies with agriculture are really designed to replace local non-renewable resources. Nevertheless, we were able to identify three situations: (1) BE-ISs with no agricultural actors, (2) BE-ISs with agricultural actors not involved in synergies with organic by-products, and (3) BE-ISs with agricultural actors involved in synergies with organic by-products.

Figure 5 shows these roles among the IS types.

In five cases, agriculture was not involved in synergies with organic by-products. This was the case when: (1) IS occurred in the context of combined heat and power production, in which forestry provided inputs for the local wood industry, the waste of which was

used as inputs for the power plant (Jyvälskylä, Kuusankoski); (2) IS occurred within the range of an agro-industrial anchor tenant that used local agricultural products (sugar beet or sugarcane) as inputs (Guitang Group, British Sugar); or (3) agriculture was involved in a synergy that did not involve organic matter—in this case, $CO_2$ (Biopark Terneuzen). In some of these cases, agriculture appeared to be a member of the symbiosis, but neither provided nor received organic by-products from other actors.

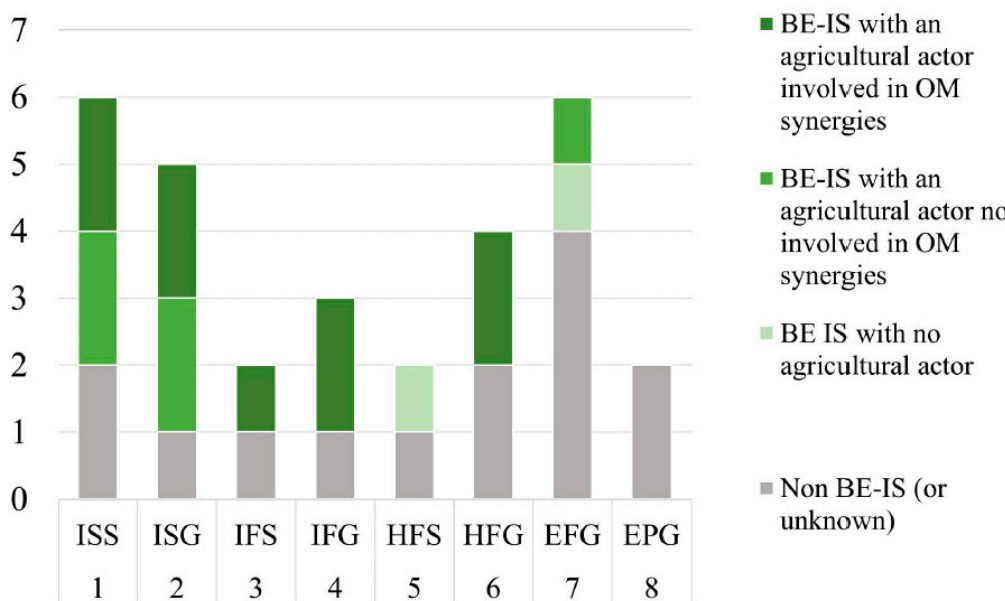

**Figure 5.** The role of agricultural actors in the selected case studies, according to the different dimensions of IS. *Y* axis: number of case studies.

When agriculture is involved in synergies, it can either provide or receive organic by-products. In three cases, agriculture provided by-products, such as crop residues (Kalundborg) or farm food waste (Relvão EIP). Agriculture also received products (7 cases), such as locally produced fertilizer (Kalundborg), treated sludge (Barceloneta), or compost (British Sugar). Finally, synergies with agriculture involved the return of nutrients to the land in less than half the BE-ISs. Agriculture was never an anchor tenant, despite the fact that, theoretically, nothing opposes such a position in BE-IS initiatives.

## 4. Discussion

### 4.1. BE among the Diversity of IS

Since our series of case studies did not correspond to ISs specifically intended to contribute to BE, our study reveals the trends that prevailed before the institutionalization of BE. Interestingly, even without the framework of BE, organic by-products were frequently involved in our case studies, and were present in most types of IS. This shows that, in some cases, both of these concepts actually pursue common goals. Despite the caution needed when drawing lessons from a limited number of cases, we identified, and can comment on, specificities of BE-ISs compared to the other ISs in our sample. BE-ISs present the same industrial diversity as other ISs. We observed a marked difference regarding anchor tenants, as they are more frequent (75% of ISs) and diversified (7 different types) in BE-ISs than in non-BE ISs (21% of ISs, only one type) (see Section 3.1). This suggests that this structure of cooperation may enhance synergies with OM by-products. It is worth noting that the most frequent sectors in which anchor tenants are present (agro-industry, chemical industry, and the energy sector) have the potential to use OM by-products either directly or indirectly. We observed no BE-ISs within a small spatial range (<5 km), which may be due

to the fact that most BE-ISs include agriculture, a spatially dispersed sector. This may also be a sign that small IS complexes, such as eco-industrial parks [106], are rarely dedicated to BE. An important distinctive dimension appears to be the emergence of symbioses, as shown in Figure 3A. The proportion of BE is high in internal emergence types (11/16 cases), moderate in hybrid emergence types (3/6), and low in ISs with external emergence (2/8). This fact suggests that external actors may tend to underestimate the BE potential within ISs. This could be explained by the traditional scope of IS, which is mainly focused on the secondary production sector [17]. Indeed, Figure 4 shows only one exchange of by-products from the agricultural sector within the same sector, although such practices have always been common in agriculture, across time and space [107]. Recycling agricultural residues in other agricultural activities appears to be generally beyond the scope of these IS case studies.

### 4.2. Contribution of Synergy Patterns to Sustainability

What we learned from our analysis of synergy patterns is that no pattern intrinsically guarantees the sustainability of BE-ISs (see Table 2). The metabolic resource and metabolic biorefinery patterns can, theoretically, help promote the BE-BE vision and increase the sustainability of local production systems. Within the metabolic resource pattern, we observed that direct energy use is important. While this may be economically advantageous, this use of OM is contrary to the principles of waste hierarchy and biomass cascading [30]. In the same pattern, the direct spreading of organic by-products enables cheap and simple nutrient recycling, but may not enable the replacement of non-renewable resources, as it may not meet local agricultural requirements (e.g., due to the timing of availability, or the nutritional composition or concentration). Similarly, biorefinery patterns tend to prioritize the most valuable uses of organic by-products, especially when links to the local metabolism are weak. The metabolic biorefinery pattern is, theoretically, the best option for an advantageous combination of BE and IS which compensates for the flaws of the two individual approaches. By transforming organic by-products, so that they meet the requirements of local metabolisms, these synergies improve the efficiency of the system. However, we observed no such advanced symbioses in our case studies, and this pattern was the least frequent among our case studies. These elements suggest that the potential contribution of BE-ISs to local sustainable biomass production is likely undervalued by both practitioners and observers. Despite the fact that BE is often spontaneously integrated into ISs by local actors (Ixx), this integration may not promote strong sustainability.

### 4.3. Designing IS to Integrate Agriculture as a Core Actor

As shown in Section 1.2, agriculture appears to be a cornerstone of sustainable BE-Iss. The examination of our selected case studies revealed the ambiguous and minor role of agriculture in current practices. Agriculture is present in most of the BE-ISs cases, and benefits from numerous synergies (Figure 4), but at the same time may not be involved in synergies (see Section 3.2). We observed that the situations in which agriculture is present, but not actually involved in synergies, mainly belong to Ixx types, and sometimes to ISx types (Figure 5). This suggests that, despite the fact that internal emergence is, theoretically, more likely to include agriculture, the synergies with residual OM are focused on the secondary sector, rather than on nutrient cycling. This shows that the apparently satisfactory incorporation of BE into classical IS case studies actually masks a more contrasted and ambiguous situation. Agriculture could be considered a global market opportunity, which could help source replacement resources outside the range of the IS. It could also be considered a convenient outlet for the disposal of by-products, as suggested by some opportunistic uses of OM in the metabolic resource pattern (see Section 3.3). The fact that agriculture never holds the role of anchor tenant shows that, so far, its synergy potential remains undervalued in IS development. One possible explanation is that agriculture is decentralized and involves many small-scale economic activities; thus, its activity is much more difficult to describe at a regional scale, as well as to mobilize and govern, than

concentrated industrial activities. Investigations of how to integrate such decentralized actors in a synergy process are still in their infancy [108].

### 4.4. Implications for Researchers and Designers

For IS researchers and designers, our observations imply that efforts should be made to consider the specific role of OM inside existing ISs, and to develop new ISs in which this role is considered able to achieve strong sustainability. This would require engaging actors that are not usually involved in ISs (i.e., farmers through delegations), and to consider the management of nutrient-rich organic material flows, if not as a priority, at least as one synergy among others. It is important to foster initiatives that explicitly aim to include these flows in regional symbiotic management, as they have not been observed to emerge either spontaneously or externally. However, it should be noted that, despite recent conceptual advances [109], tools for the design and evaluation of the technical performance of OM synergies are still lacking. This stems from the fact that BE embeds multiple and intricate stakes, which raises complex epistemological questions [15].

### 5. Conclusions

Our analysis of 30 case studies shows that industrial symbioses (ISs) already contribute, to some extent, to the bioeconomy (BE). We observed that the characteristics of some industrial symbioses including bioeconomy (BE-ISs), differ markedly from the characteristics of other ISs, namely in their range and the presence of anchor tenants. Despite this trend, the synergy patterns we identified reveal that the contributions of IS to BE does not guarantee a path towards strong sustainability. Our results also suggest that IS practitioners undervalue both the potential of BE and the need to integrate agriculture into ISs as an activity of high synergistic potential. Enhancing sustainability through BE-ISs requires a change in the perception of IS, to ensure that equal attention is paid to synergies within the primary sector, as opposed to the traditional focus on the secondary sector.

Our observations allowed us to make some recommendations for researchers and practitioners willing to foster sustainability through the development of BE-ISs. In this case, we recommend (1) broadening the general IS framework to pay more attention to agricultural actors and their synergistic potential for sustainable biomass production, (2) dedicating particular efforts to initiating ISs that include local nutrient recycling for agriculture, and (3) studying the specific challenges associated with this new form of IS. These challenges include, for instance, the involvement of decentralized agriculture actors such that they assume a role similar to an anchor tenant; selection of a spatial and sectoral perimeter that favors the involvement of agriculture; the initiation of such synergies; and building trust between agricultural actors and other traditional IS actors. Our work also underlines the importance of further research on how an industrial symbiosis, dedicated to sustainable bioeconomy, can be initiated in an area with known synergistic potential. Knowledge of common synergy patterns is crucial to lay the foundations for a new type of ISs, as a plausible promise to engage actors in collaborative action [110].

**Author Contributions:** Conceptualization, N.B.; methodology, N.B.; validation, T.W. and G.J.; investigation, N.B.; resources, T.W. and G.J.; data curation, N.B.; writing—original draft preparation, N.B.; writing—review and editing, T.W., G.J. and M.D.; supervision, T.W., G.J. and M.D.; project administration, T.W. All authors have read and agreed to the published version of the manuscript.

**Funding:** This research is supported by Veolia and the French National Association for Research & Technology (ANRT) with an Industrial Agreement of Training through Research (CIFRE contract).

**Institutional Review Board Statement:** Not applicable.

**Informed Consent Statement:** Not applicable.

**Data Availability Statement:** The data presented in this study are available on request from the corresponding author.

**Acknowledgments:** The authors would like to acknowledge Veolia Recherche et Innovation (VERI) for their support.

**Conflicts of Interest:** The funders had no role in the design of the study, the collection, analyses, or interpretation of data, the writing of the manuscript, or the decision to publish the results.

## Appendix A

*Appendix A.1. Assessment of Industrial Symbiosis Characteristics*

### Appendix A.1.1. Industrial Diversity

The number of different activity sectors involved in synergies was assessed using NACE typology [102], aggregated according to the SNC/CITI Ax38 classification. This typology uses 38 categories to differentiate all economic activities. We considered this to constitute an interesting level of aggregation, to assess the "shared norms" [111] between actors. Categories correspond to letters or a combination of letters. For each case study, we associated each actor involved with its NACE reference code, and its equivalent within the Ax38 aggregation. We assessed the industrial diversity by counting the number of different industrial sectors, which correspond to categories from A to H.

### Appendix A.1.2. General Network Structure

The notion of anchor tenant structure [60] provides interesting information about ISs. It underlines the existence of a central actor that is able to exchange diverse material with many other actors, either by accepting by-products or having by-products accepted by other actors. We evaluated this characteristic according to the previous definition, and by the analysis of the network structure used in the studies. Using NACE typology [102], we noted the activity sector of the anchor tenant, when present.

### Appendix A.1.3. Spatial Range

We defined spatial range as the greater geographical distance between two actors in an IS. This indication is rarely provided in the literature. When it was not explicitly stated, we estimated this range using all the information in our possession, including online maps. The estimate is only a proxy of spatial range, which is why we chose to describe the range with spatial intervals. We used four intervals: (i) small—0 to 5 km; (ii) medium—5 to 20 km; (iii) large—20 to 50 km, and (iv) very large—over 50 km.

*Appendix A.2. Assessment of Bioeconomic Characteristics*

### Appendix A.2.1. BE-IS

Among the selected case studies, we identified industrial symbioses including bioeconomy (BE-ISs) as ISs that included at least one exchange of organic by-product. Using this subset of case studies, we performed additional analyses.

### Appendix A.2.2. OM Actors of BE-IS

We used NACE typology [102] to identify the sectors involved in the exchange of organic by-products in BE-ISs. Based on the actor categorization, we enumerated, for each case study, the known synergies involving OM, and noted the NACE category of both the producer and receiver of each exchange. We also analyzed the use of OM within these synergies. Due to its important role regarding sustainable biomass production, we decided to focus specifically on agricultural actors in BE-ISs. We described the specific interactions of these actors with other the actors inside BE-ISs.

## Appendix B

*Appendix B.1. Typology Based on Three Constitutive Dimensions*

IS corresponds to a high diversity of different situations. Capturing this diversity with a robust and consistent framework is a prerequisite for analyzing the presence of BE

within this range of different practices. Chertow (2000) [18] proposed the first distinction between different IS types, based on the scale of the synergy network and the nature of the synergies involved. Following this seminal work, authors have analyzed specific characteristics of IS and established unidimensional descriptions. The most acknowledged distinction is between self-organized and planned ISs [103]. However, its relevance tends to decrease with the increasing diversity of case studies. Some multidimensional descriptions of ISs have also been proposed. Golev & Corder [103] introduced a description system to evaluate the performance of ISs in reaching environmental or economic goals, allowing comparisons to be made between different case studies. Cerceau et al. [112] analyzed the relation between the geographical and temporal dimensions of industrial ecology initiatives in port cities. The advantage of such typologies is that they allow the differentiation and comparison of ISs according to accurate criteria. They provide precious knowledge of different types of ISs according to specific dimensions. Lately, a significant contribution was made by Boons et al. [62], through a typology of ISs intended to overcome what the authors call the "*problem of equivalence*". This typology considers IS a dynamic process, and the authors introduce a generic sequence of events to compare different case studies. They identified seven distinct IS types, which are (1) self-organization, (2) organizational boundary change, (3) facilitation—brokerage, (4) facilitation—collective learning, (5) pilot facilitation and dissemination, (6) government planning, and (7) eco-cluster development. Although this work remains the most comprehensive to date, no consensus on IS typology has emerged, as we can observe that recent IS reviews [17,113] do not specifically refer to this typology—nor do they propose a new one. Moreover, this typology does not rely on clearly formulated criteria, which makes it difficult to use as a comparative tool. These are the reasons why we introduce new descriptive characteristics/features in the Material and Methods section of our study.

Appendix B.1.1. Symbioses Emergence

IS literature has long studied the emergence of initiatives, and makes a distinction between bottom-up cases, such as the historical example of Kalundborg [64], and top-down cases in which IS is pushed by an external policy. A middle-out approach was also described in some studies [93]. The important question, addressed by the emergence dimension, is "Who is the initiator of the IS?". Relevant knowledge includes whether the IS emerged by itself, with internal co-located actors, or if it was induced by external actors. As we describe further, this dimension must be separated from the governance process that takes place after IS emergence. We propose new terms to explicitly refer to the emergence process, minimizing the risk of confusion with governance. Emergence of IS can be driven by (i) internal actors, (ii) external actors, or (iii) both internal and external actors.

Internal emergence: The IS emerges as actors spontaneously engage in synergies with each other. It begins with the cooperative behavior of individual actors. The actors, or groups of actors, that promote IS are directly involved in the synergies. It can be described as actor-driven, and refers to the traditionally-named bottom-up approach.

External emergence: The IS is initially a project promoted, suggested, or imposed by a third party, external to the symbiosis. This is, for instance, the case when public or academic actors drive SI dynamics through facilitation actions. We can call this type of IS policy-driven, or top-down symbioses.

Hybrid emergence: As suggested by Costa & Ferrão [93], there might be significant contribution to the emergence of an IS from both internal and external actors. This refers to the middle-out situation, which we include as a midpoint between internal and external emergence.

Appendix B.1.2. Governance for Synergy Design

It is often difficult to distinguish, in literature, the description of the emergence of the symbiosis and the governance that takes place once actors in synergy. The terms self-organized, facilitated, and planned, formalized by Chertow [103], sometimes refer to the

course of the IS [104], and sometimes to the emergence process [86]. The meaning of those terms appears, then, to be similar to the meaning of, respectively, the terms bottom-up, middle-out, and top-down. However, the process of emergence can be distinguished from the governance of the synergies once the IS exists. The question raised by the governance dimension is "How are synergies between actors designed and agreed upon?". The determining factor is the level of participation of actors in the synergy design. We propose the use of existing terminology, with a cautious definition of each term. Governance dimensions include, then, (i) self-organized, (ii) facilitated, and (iii) planned synergies.

Self-organized synergies: Synergies are bilateral contracts between actors, and fairly independent from each other. This corresponds to the maximal level of autonomy of the actors.

Facilitated synergies: Synergies are designed in a collective process involving the actors. The process can be facilitated by an internal or an external actor, and can involve the other actors to varying degrees. Facilitation actually refers to an extremely large range of practices, and the definition that we use for facilitation does not address this problem. This stems from the fact that there are no significant investigations in the literature that describe the different forms of facilitation. As this was not the purpose of our study, we cannot increase the accuracy of this term.

Planned synergies: The word 'planned' can evoke the planning of the IS (which is related to the emergence), but should here be understood as the planning of synergies. In planned ISs, synergies are designed by a third party, without involving the actors. Flow exchanges and technical solutions are analyzed separately from the actors. It encompasses the 'engineered' synergies [114] in the greenfield development of Eco-Industrial Parks. Extreme cases of planned ISs are rare, because the participation of actors, to at least some degree, is a prerequisite to engaging in synergies. The traditional distinction between self-organized and planned ISs is, then, of limited relevance, since the majority of cases that are not self-organized are facilitated, rather than planned.

Appendix B.1.3. Governance for Synergy Design

Paquin & Howard-Grenville [105] coined the term '*serendipity*' to designate the ability of actors to build new synergies independently from overarching goals, in opposition to goal-directed ISs. Baas & Boons [78] also suggested this idea, distinguishing epistemic rationale, which corresponds to a "narrow economic/rational focus", from adaptation rationale, which "allows a slow and cautious process by the intermediary organization". Contrary to what might be intuitively thought, this dimension of IS cannot be directly inferred from other dimensions. For instance, facilitated ISs, which one might think of as necessarily goal-directed, can either foster or restrain the serendipity of actors. The important question to address is "What are the synergies for?". New synergies can, in fact, serve either individual or collective interests. The distinction depends on what is considered a collective interest. In this work, we chose to consider implicit collective goals to be a goal-directed strategy. Such goals might not be loudly advocated by coordinating structures, but rather represent a shared vision of the situation, which strongly frames the quest for synergies and their purpose. These implicit goals can play a determinant role in ISs, and are generally the consequence of contextual situations or actions of external actors. We used the terms from Paquin & Howard-Grenville [105], and kept similar definitions.

Serendipitous IS: The synergies pursue actors' individual goals. The synergies are mainly guided by economic market dynamics and incentives.

Goal-Directed IS: The synergies pursue explicit or implicit collective goals. These goals can be given by a coordinating structure, or by the general context. Synergies can, for instance, be constrained by regulation, encouraged by a political action plan, or designed in a facilitation process.

## Appendix C. Extensive Description of Case Studies

**Table A1.** Selection and main characteristics of case studies. Country names are presented according to ISO 3166. Continent abbreviations: AS, Asia; EU, Europe; NA, North America; OC, Oceania; SA, South America. Corpus 1, Boons et al., 2016; Corpus 2, Mortensen and Kørnøv, 2019. Emergence, Governance, and Serendipity are defined in Section B. ID: Industrial Diversity as defined in Section 2.2.1. AT, Anchor Tenant, as defined in Section 2.2.1. NA, Not Assessable.

| Name, Country, Continent | Corpus | Emergence | Governance | Serendipity | Type of Activities (SCN A38) | ID | AT | AT Sector | BE-IS (OM-Synergies) | Range | Refs. |
|---|---|---|---|---|---|---|---|---|---|---|---|
| Kalundborg, DK, Eu | 1 | Internal | Self-organized | Goal-Directed | A, CD, CE, CF, CG, D, O | 6 | Yes | D CD | Yes | Medium | 1,2,3 |
| Styria, AT, Eu | 1 | Internal | Self-organized | Serendipitous | A, CB, CC, CD, CE, CG, CH, D, E, O | 9 | No | - | Yes | Very Large | 4 |
| Guayama, PR, NA | 1 | Internal | Self-organized | Goal-Directed | CD, CF, CG, CH, CM, D | 6 | Yes | D | No | Small | 5 |
| Kwinana, AU, Oc | 1 | Internal | Facilitated | Goal-Directed | A, CD, CE, CG, CH, D; E, H | 8 | No | - | Yes | Medium | 6,7 |
| Gladstone, AU, Oc | 1 | Internal | Facilitated | Serendipitous | B, CE, CG, CH, D, H | 6 | No | - | No | Medium | 7 |
| Nanjangud, IN, As | 1 | Internal | Self-organized | Serendipitous | A, CA, CD, CE, CG | 5 | Yes | CA | Yes | Large | 8 |
| Jyvälskylä, FI, Eu | 1 | Internal | Self-organized | Goal-Directed | A, CC, D, F, O | 4 | Yes | D | Yes | Medium | 9 |
| Kuusankoski (KYMI EIP), FI, Eu | 1 | Internal | Self-organized | Serendipitous | A, CC, CE, D, E | 5 | Yes | CC | Yes | Medium | 10, 11 |
| Guitang Group, CH, As | 1 | Internal | Self-organized | Goal-Directed | A, CA, CC, CE, CG | 5 | Yes | CA | Yes | Medium | 12,13 |
| British Sugar, GB, Eu | 1 | Internal | Self-organized | Serendipitous | A, CA, CE, CG, D | 5 | Yes | CA | Yes | Large | 14 |
| Humber Industrial Symbiosis Program, GB, Eu | 1 | Hybrid | Facilitated | Goal-Directed | A, CA, CD, CG, CM, D, E | 7 | No | - | Yes | Large | 15 |
| West-Midlands Insdustrial Symbiosis Program, GB, Eu | 1 | Hybrid | Facilitated | Serendipitous | CA, CE, CG, CL, CM, D, I, MB, O, R | 6 | Yes | CG | Yes | Large | 15 |
| Rotterdam Harbour and Industry Complex, NL, Eu | 1 | Internal | Facilitated | Goal-Directed | CD, CE, CG, CH, D, O | 5 | No | - | No | Large | 16,17 |
| Ulsan, KR, As | 1 | External | Facilitated | Goal-Directed | CC, CD, CE, CG, CH, CL, D, E, O | 8 | No | - | No | Medium | 18, 19 |
| Tianjin Economic-Technological Development Area (TEDA), CH, As | 1 | External | Facilitated | Goal-Directed | CE, CH, CI, CL, D, E | 6 | No | - | No | Large | 20, 21 |

**Table A1.** *Cont.*

| Name, Country, Continent | Corpus | Emergence | Governance | Serendipity | Type of Activities (SCN A38) | ID | AT | AT Sector | BE-IS (OM-Synergies) | Range | Refs. |
|---|---|---|---|---|---|---|---|---|---|---|---|
| Fort Devens Army Base, US, NA. | 1 | External | Facilitated | Goal-Directed | CC, CE, CF, CI, CK, CM, E, H, I, JA, MC, O, P, R | 8 | No | - | No | Small | 22 |
| Händelö island, SE, Eu | 1 | Internal | Facilitated | Serendipitous | A, CE, D, E, O | 4 | Yes | D | Yes | Medium | 23, 24 |
| Biopark Terneuzen, NL, Eu | 1 | External | Facilitated | Goal-Directed | A, CA, CD, CE, D, E, H | 7 | No | - | Yes | Medium | 25 |
| Zhejiang Hangzhou Bay Shangyu Industrial Area (SYIA), CN, As | 1 | Internal | Self-organized | Serendipitous | CE, D, E | 3 | No | - | No | Medium | 26, 27 |
| Campbell Industrial Park (Hawaï), US | 1 | Internal | Self-organized | Serendipitous | B, CD, CG, D, E, R | 5 | Yes | D | No | Medium | 28 |
| Porto Marghera, IT, Eu | 2 | Hybrid | Facilitated | Goal-Directed | CD, CE, D, E | 4 | No | - | No | Small | 29 |
| Industrial Symbiosis Platform (Sicily), IT, Eu | 2 | Hybrid | Facilitated | Serendipitous | A, CA, CC, CE, CG, CH, CI, CJ, CK, CM, D, E, F, G, H, M, N, Q, S | 15 | No | - | Unknown | Very Large | 30 |
| Relvao Eco Industrial Park, PT, Eu | 2 | Hybrid | Facilitated | Goal-Directed | A, CC, CE, E | 4 | Yes | E | Yes | Medium | 31 |
| Barceloneta, PR, NA | 2 | Internal | Self-organized | Goal-Directed | A, CA, CE, CF, D, E | 6 | Yes | CF | Yes | Large | 32 |
| Northern Region Industrial Estate (NRIE), TH, As | 2 | External | Planned | Goal-Directed | CA, CK, CI, CM | 4 | No | - | Unknown | Small | 33 |
| Santa Cruz EIP, BR, SA | 2 | External | Facilitated | Goal-Directed | CE, CG, CH, CM, D, E, N | 6 | No | - | No | Medium | 34,35 |
| Paracambi, BR, SA | 2 | External | Planned | Goal-Directed | CB, CE, CG, CH, CJ, D, E | 7 | yes | D | No | Large | 34 |
| Bazancourt-Pomacles, FR, Eu | 2 | Internal | Facilitated | Goal-Directed | A, CA, CE, H, M, O, P | 4 | yes | CE | Yes | Medium | 36 |
| Kawasaki, JP, As | 2 | External | Facilitated | Goal-Directed | CC, CE, CG, CH, E | 5 | yes | CG | Yes | Medium | 37, 38 |
| Deux Synthe, FR, Eu | 2 | Hybrid | Facilitated | Goal-Directed | A, CE, CG, CH, D, E, F | 7 | No | - | No | Medium | 39 |

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
