# Peer review of "Towards a Sustainable Bioeconomy through Industrial Symbiosis: Current Situation and Perspectives"

_sustainability, doi:10.3390/su14031605_

Round 1

Reviewer 1 Report

I am honoured to have had the opportunity to read this work and I offer the following comments to the authors for further improve the article.

  • Section 1.2.1: I suggest an additional shorter theoretical chapter to show how the bioeconomy is embedded in industrial symbiosis activities as one of the tools of the circular economy (for an example of the presentation of industrial symbiosis as a tool of the circular economy through a case study, the authors can refer to this article: https://hrcak.srce.hr/file/307136).
  • Section 2.1: I suggest a more detailed presentation of the background to the selection of case studies - at least which bibliographicc databases were used for relevant articles.
  • Figure 3, D1: The two figures overlap.
  • Conclusion: In conclusion, the authors' further work in this area should be presented - with an emphasis on how this research will help them.

Reviewer 2 Report

In this paper, the authors define Industrial Symbioses including Bioeconomy (BE-IS) as ISs that include at least one exchange of organic by-products, thus assessing the extent to which it can contribute to a strong sustainability. Here are many unreasonable presentations such as structure, logic that need to be addressed. Please see specific comments below:

1.Please improve the graphics quality. Please review the figures and revise accordingly.

  1. Would you explicitly specify the novelty of your work? What progress against the most recent state-of-the-art similar studies was made? The following paper can be analyzed.

1)Production capacity assessment and carbon reduction of industrial processes based on novel radial basis function integrating multi-dimensional scaling. Sustainable Energy Technologies and Assessments

2Energy consumption analysis and saving of buildings based on static and dynamic input-output models. Energy

  1. Write down managerial implications and limitations.

4.Conclusions are not up to mark; revise them.

  1. Overall, there are lack of detailed explanation on the variation of findings.

Round 2

Reviewer 2 Report

The auhors should be answer these questions of the reviewers in detail.

Author Response

Dear reviewer, 

We strongly apologize that the last version of our manuscript was unwillingly flawed due to mistakes that occured during the saving of our file. We unfortunately noticed the mistake only after reading your report. 

Please, find our new version of the manuscript, that has been thoroughly checked before sending. It correspond to the file that you should have recieved last time, after our first revision. Please, note that this second revision only includes corrections to the layout problems. You will find enclosed a new copy of the detailed aswers to your comments during the first review. We are sorry again for the trouble and hope that you will find our corrections fitted.

Sincerely yours, 

Nicolas Bijon, in the behalf of all the authors

Round 3

Reviewer 2 Report

The related word should be added in the accepted paper. Thanks.